# Magnitude of *Mycobacterium tuberculosis*, drug resistance and associated factors among presumptive tuberculosis patients at St. Paul's Hospital Millennium Medical College, Addis Ababa, Ethiopia

**Melkayehu Kassa**[1], **Kassu Desta**[2], **Rozina Ambachew**[1], **Zenebe Gebreyohannes**[1], **Alganesh Gebreyohanns**[1], **Nuhamen Zena**[1], **Misikir Amare**[3], **Betselot Zerihun**[3], **Melak Getu**[3], **Addisu Gize**[1]*

1 St. Paul's Hospital Millennium Medical College, Addis Ababa, Ethiopia, 2 College of Health Science, Addis Ababa University, Addis Ababa, Ethiopia, 3 Ethiopian Public Health Institute, Addis Ababa, Ethiopia

* addisu.gize@sphmmc.edu.et, konjoaddisu@gmail.com

## Abstract

### Background

*Mycobacterium tuberculosis* (*M. tuberculosis*) remains one of the most significant causes of death and a major public health problem in the community. As a result, the aim of this study was to determine magnitude of *Mycobacterium tuberculosis*, its drug resistance, and associated factors among presumptive tuberculosis (TB) patients at St. Paul's Hospital Millennium Medical College, Addis Ababa, Ethiopia.

### Methods

Cross-sectional study was conducted at St. Paul's Hospital Millennium Medical College (SPHMMC), Addis Ababa, Ethiopia from January to July 2019. Demographic and clinical data were collected by structured questionnaire through face to face interview. Using microscopic examination and GeneXpert MTB/RIF assay and culturing in the Lowenstein-Jensen (LJ) culture media, we collected and analyzed both pulmonary and extra-pulmonary clinical samples. Data were analyzed by SPSS version 23. Binary logistic regression was done to identify the associated risk factors and p-value less than 0.05 was taken as significant association.

### Results

Of the total 436 respondents, 223(51%) were male. The mean ±SD age of the participants was 38±17years. Overall, 27/436(6.2%) of the participants had confirmed *Mycobacterium tuberculosis* using the GeneXpert MTB/RIF assay and LJ culture media, and two isolates were resistant to RIF and one to INH medication, with two (0.5%) being MDR-TB. MTB infection was associated with previous TB contact history, patient weight loss, and CD4+ T-cell counts of 200-350/mm3 of blood.

**Data Availability Statement:** All data relevant to this study are available on the manuscript.

**Funding:** This research work was supported by Addis Ababa University, Ethiopia. The funders had no role in study design, data collection and analysis, decision to publish, or preparation of the manuscript.

**Competing interests:** The authors have declared that no competing interests exist.

**Abbreviations:** ATCC, American Type Culture Collection; BCG, Bacillus Calmette–Guérin; DOT, Directly Observed Therapy; DST, Drug Susceptibility Test; FMOH, Federal Ministry of Health; HIV, Human Immunodeficiency Virus; INH, Isoniazid; IQC, Internal Quality Controls; MDR-TB, Multidrug-Resistant Tuberculosis; MOTT, Mycobacteria other than TB; MTB, *Mycobacterium Tuberculosis*; OADC, Oleic Acid Albumin Dextrose Complex; PPE, Personal protective equipment; PTB, Pulmonary Tuberculosis; RIF, Rifampicin; RMR, Rifampicin Mono-Resistant; SOP, Standard Operating Procedures; SPHMMC, Saint Paul's Hospital Millennium Medical College; SR, sample reagent; STM, Streptomycin; TB, Tuberculosis; TTD, Time to detection; WHO, World Health Organization.

## Conclusion

The magnitude of *M. tuberculosis* and MDR-TB in this study underscores the need for improved early case detection and management of MDR-TB in order to reduce transmission and patient suffering.

## Background

Tuberculosis (TB) is a disease caused by *Mycobacterium tuberculosis* complex bacteria. The most common mode of transmission is through the respiratory system when an infected person coughs or sneezes near healthy people. However, *Mycobacterium bovis* can infect us while we are consuming and drinking unboiled milk from infected cattle [1]. World Health Organization (WHO) estimated that 9.9 million people developed tuberculosis (TB) and 1.5 million died of TB globally in 2020 and 22% of the world population are infected with latent *Mycobacterium tuberculosis*, of which 95% cases occur in resource limited settings [2]. According to a WHO report published in 2019, estimates that 3.4% of new patients, and 18% of previously treated cases have MDR-TB [3].

According to this report, the incidence of tuberculosis in Ethiopia is estimated to be 247 per 100,000, placing the country seventh in the world and fourth among Sub-Saharan African countries with significant TB burdens [3, 4]. In one study which was conducted in Ethiopia in 2018, the prevalence of MDR-TB in Ethiopia was 7.24%, of which 2.1% was new cases and 21.1% relapse cases [4].

Studies conducted in Ethiopia showed higher mortality rate in different health institutions; 11.3% patients died in Mekelle Hospital and Ayder Comprehensive Hospital [5], 14% children with TB and HIV co-infected from University of Gondar Comprehensive Specialized Hospital [6], and 29.5% of the patients died from MDR-TB in different hospitals of Amhara region, Northwest Ethiopia [7]. In general, the TB mortality rate in Ethiopia decreased from 393.8/100,000 to 100/100,000 between 1990 and 2016 with 75% decline, indicating a slow decline. As a result, study suggested that males had a higher TB mortality rate than females [8]. Microscopic examination in sputum stained smear detection rate for *M. tuberculosis* using light microscope up 80% in the case of fluorescence method applications ranged from 20% to 80% respectively.

This method can be used to detect TB when the clinical sample contains sufficient tuberculosis micro-organism without testing of the resistance pattern, but with less detection rate in immunocompromised people like HIV infected individuals and children, since they could not produce enough sputum [9].

Hence, GeneXpert MTB/RIF assay test should be used as an initial diagnostic test for TB and rifampicin resistance detection in patients suspected of having TB, MDR-TB or HIV-associated TB, as this test has high sensitivity and specificity [10]. Therefore, the aim of this study was to determine magnitude of *Mycobacterium tuberculosis*, magnitude of drug resistance and associated factors among presumptive TB patients referred to St. Paul's Hospital Millennium Medical College, Addis Ababa, Ethiopia.

## Materials and methods

### Study area

The study was conducted in St Paul's Hospital Millennium Medical College, Addis Ababa, Ethiopia. The hospital serves patients from all over the country. It has 392 beds, with

catchment population of more than 5 million. On average, the microbiology laboratory receives and process five sputum samples for pulmonary tuberculosis and two extra-pulmonary tuberculosis clinical samples per day.

### Study design and period

From January to July 2019, a cross-sectional study was conducted at SPHMMC in Addis Ababa, Ethiopia. All patients who visited the hospital were the source population, and all patients with the diagnosis of presumptive pulmonary tuberculosis who visited the microbiology laboratory and met the inclusion criteria were the study population.

### Inclusion and exclusion criteria

All patients with a presumptive diagnosis of *Mycobacterium tuberculosis* who visited a microbiology lab were included in the study. Patients with insufficient specimens or a history of known *Mycobacterium tuberculosis* treatment resistance were excluded from the study.

### Variables

Magnitude of *Mycobacterium tuberculosis* and its drug resistance pattern among presumptive patients were dependent variables. Whereas, socio-demographic characteristics, possible risk factors like; TB contact history, previous treatment for TB, presumptive Drug Resistance Tuberculosis (DR-TB), BCG vaccination status, $CD4^+$ and HIV viral load counts were independent variables.

### Sample size and sampling technique

To get the maximum sample size, the sample size was determined using the assumption of a single population proportion formula, taking into account a proportion of 50%, a margin of error of 5%, and a confidence level of 95%. The calculation result was as follows:

$$n = \frac{\left(z^{\alpha}/_2\right)^2 \ p\,(1-p)}{d^2} \Rightarrow \frac{(1.96)^2 \ 0.5(1-0.5)}{(0.05)^2} = 384 \ \text{study subjects.}$$

Where: n = minimum sample size,
P = estimated proportion of patients with *Mycobacterium tuberculosis* for the study population, and taking 10% non-response rate, the final sample size become 422 participants.
d = the margin of sample error, $z^{\alpha}/_2$ = the standard normal variable at $1\text{-}^{\alpha}/_2$ confidence level and we used consecutive sampling technique was used to select the study population.

### Data collection procedure

Data collectors were given training and instructions on how to collect the information. The study participants' socio-demographic status and related risk variables were collected using a structured questionnaire. For each patient with a presumptive diagnosis of *Mycobacterium tuberculosis*, a 2–4 ml of clinical sputum, lymph node aspirate or peritoneal and pleural fluid, and gastric aspirate samples were collected. For children who were unable to cough up sputum, we used gastric aspirate or induced sputum with physician assistance.

### Laboratory procedures

For all samples, GeneXpert MTB/RIF, microscopy, and culture tests were done in parallel.

In the case of sputum samples, pellets were used for GeneXpert MTB/RIF assay. Sample reagent (1.5 ml) was added to 0.5 ml of the re-suspended sputum pellet and manually agitated twice at room temperature during a 15-minutes period.

In the case of other clinical body fluids, we transferred the entire specimen to a conical centrifuge tube, and concentrate the specimen at 3000 g for 15 minutes, carefully poured off the supernatant through a funnel into a discard can containing 5% sodium hypochlorite, re-suspend the deposit to a final volume of 2 ml by Phosphate Buffer Saline (PBS), using transfer pipette, we added a double volume of the GeneXpert MTB/RIF sample Reagent (1.4 ml) to 0.7 ml (2:1) of suspension, then 2 ml transferred to the concentrate and load to the GeneXpert MTB/RIF cartridge. The GeneXpert MTB/RIF purifies *M. tuberculosis* from these clinical samples. The genetic content of *M. tuberculosis* captured and subsequently amplifies the genomic DNA by polymerase chain reaction (PCR). In addition, it identifies RIF's resistance mutations in the RNA polymerase beta (rpoB) gene of *M. tuberculosis* in all clinically important samples within 2 hours [11].

Regarding microscopic examination, all sputum smears are prepared from decontaminated and concentrated specimens. The smears stained with Ziehl-Neelsen (ZN) staining techniques, could be used to count both viable and non-viable bacilli as acid-fast bacilli (AFB).

This method uses a carbolfuchsin as primary stain, acid alcohol as decolorizer, and methylene blue as counterstain. Acid-fast organisms stain red, while the background of debris stains blue. The ZN stain confirms the acid-fast property of mycobacteria using microscopy examination. Bacillary density will be graded as scanty, 1+, 2+, and 3+, and all such smears will be defined as "smear-positive".

Lowenstein-Jensen (LJ) culture medium was used which incorporates congo red and malachite green to inhibit unwanted bacteria for culturing. Once good growth was obtained, the positive slants were stored in a cool, dark place to archive the positive *M. tuberculosis* isolates.

## Data quality assurance

The questionnaire was pre-tested and proper training was given for data collectors. The quality of data was maintained following the pre-analytical, analytical and post-analytical steps through each day supervision using standard laboratory procedures (SOPs).

## Data analysis and interpretation

The collected data were entered to EPI info 2002 version 3.32 after data cleaning it was exported to SPSS version 23 windows software computer program for analysis. The logistic regression was employed to assess the association between TB and its risk factors. A p-value of less than 0.05 was considered as statistical significance.

## Ethical considerations

This study was approved by Department of Medical Laboratory Science, College of Health Sciences, Addis Ababa University, Addis Ababa, Ethiopia. IRB also obtained from St. Paul's Hospital Millennium Medical College, Addis Ababa, Ethiopia. Written informed consent was secured from each participant greater than 18 years old and assents were obtained for those less than 18 years old. Infected patients and/or those who had resistance *M. tuberculosis* were informed to their health care provider for better care and management.

## Operational definition

MDR-TB: is non-susceptibility of *M. tuberculosis* at least two first line TB drugs (isoniazid and rifampicin).

Presumptive TB: a patient who presents with symptoms or signs suggestive of TB including cough >2week, fever >2week, significant weight loss, haemoptysis, any abnormality chest radiograph.

Presumptive MDR-TB: smear positive previously treated patients who define as relapse, return after default, and failure; new smear positive pulmonary TB patients who sputum remains smear positive at month 2 or 3 of treatment.

## Results

### Socio-demographic characteristics

A total of 436 participants were enrolled during the study period, of this 223 (51%) were male. The mean ± SD age the participants were 38±17years. The highest age range was 35–49 years old, while the youngest was under 15 years old. Majority of the respondents were 240 (55%) urban residents, and 214 (49%) had monthly income of 100–1000 Ethiopian Birr (Table 1).

**Table 1. Socio-demographic characteristics among presumptive TB patients at SPHMMC, Addis Ababa, Ethiopia, 2019.**

| Variables/ characteristics | | No. of Participants | Percentages (%) |
|---|---|---|---|
| Sex | Male | 224 | 51 |
| | Female | 212 | 49 |
| Age groups | <15 years | 39 | 9 |
| | 15–24 years | 56 | 13 |
| | 25–34 years | 98 | 22 |
| | 35–49 years | 127 | 29 |
| | >50 years | 116 | 27 |
| Residence | Urban | 240 | 55 |
| | Rural | 196 | 45 |
| Family size/house | 1–3 | 152 | 35 |
| | 4–6 | 220 | 50 |
| | >6 | 64 | 15 |
| Marital status | Single | 146 | 33 |
| | married | 238 | 55 |
| | Divorced | 20 | 5 |
| | Widowed | 32 | 7 |
| Occupational status | Laborer | 97 | 22 |
| | Government workers | 97 | 22 |
| | Private workers | 109 | 25 |
| | House wife | 70 | 16 |
| | Student | 63 | 15 |
| Educational status | No formal Education | 119 | 27 |
| | 1-8th grades | 147 | 34 |
| | 9-12th grades | 106 | 24 |
| | >12th grade | 64 | 15 |
| Monthly Income | <100 Birr | 60 | 14 |
| | 100–1000 Birr | 83 | 19 |
| | 1001–2000 Birr | 155 | 35. |
| | 2001–3000 Birr | 59 | 14 |
| | 3001–4000 Birr | 32 | 7 |
| | 4001–5000 Birr | 25 | 6 |
| | >5001 Birr | 22 | 5.0 |

### Clinical data

There were a total of 374 (85.8%) pulmonary tuberculosis presumptive patients and 62 (14.2%) extra-pulmonary tuberculosis presumptive patients, with 130 (30%) of them being HIV positive. Presumptive TB was discovered in 422 (96.8%) of the individuals, while presumptive DRTB was found in 14 (3.2%) of the total participants.

In this study 33(7.5%) of the participants had history of contact with a TB patient, 68 (15.6%) had history of alcohol drinking, and 22 (5%) were cigarette smokers. Among the study participants 319 (73.1%) had fever, 311 (71.3%) had night sweating, and 365(83.7%) had cough. Out of 130 HIV positive participants, 104 (81%) were on antiretroviral (ART) treatment and were monitored based on their $CD4^+$ T-cells count. In addition, 119 (91.5%) participants were tested for HIV viral load. Higher magnitude *M. tuberculosis* seems to be appeared for those who have $CD4^+$ count 200-350/$mm^3$ (5/34) and viral load $\geq$1000/$mm^3$ (6/90) (Table 2).

### Magnitude of *M. tuberculosis* and its resistance pattern

Out of the total participants of clinical specimen, *M. tuberculosis* was detected in 36(8.3%) of the samples with GeneXpert MTB/RIF, and of which only 2 (0.5%) of them were RIF resistant. Regarding culture result, 27(6.2%) of the participants were confirmed for M. tuberculosis infection and one *M. tuberculosis* strain was resistant for Isozianide drug (mono-resistant) and 2 were resistant for Isozianide and RIF (Multidrug resistant TB).

### Factors associated to *M. tuberculosis*

The bivariate logistic regression analysis of socio-demographic characteristics revealed that participants under the age of 15 years old were 1.8 times (95% CI: 0.4, 8.1) more likely to develop M. tuberculosis than those over 50 years old. Widowed participants were 2.6 times (95% CI: 0.4, 17) more likely to have *M. tuberculosis* than unmarried ones, and government workers were 1.8 times (95% CI: 0.6, 5.9) higher than housewives (Table 3). On bivariate logistic analysis, contact history with tuberculosis-infected patients, pneumonia confirmed by chest X-ray examination, and $CD4^+$ results were associated factors for *M. Tuberculosis*; however, none of these factors were associated in the multivariate analysis (Table 4).

## Discussion

People aged 35 to 49 years old, as well as those living in families with 4–6 members, had the highest frequency of tuberculosis. In terms of occupation, laborers earning between 1000 and 2000 Ethiopian Birr per month were the most susceptible to tuberculosis. This could be because these age groups are more likely to be subjected to high workloads and have a greater range of motion.

The current study found that as the number of people living together increases (5–6 family size), *M. tuberculosis* positivity increases as well. Other studies have found that having a larger family and malnutrition contribute to the development of tuberculosis [12], however, the present study found no association between family size/household and *M. tuberculosis*.

Higher *M. tuberculosis* was detected among participants in presumptive diagnosis of tuberculosis 25/436 (5.7%), non-vaccinated for BCG 18/436 (4.1%) than vaccinated, non-alcoholic drinkers 21/436 (4.8%) than alcoholic drinkers, and non-cigarette smokers 25/436 (5.7%) than smokers.

Again, among symptomatic tuberculosis patients, higher *M. tuberculosis* results were observed in those with night sweating 23/436 (5.2%), fever 22/436 (5.0%), weight loss 20/436

**Table 2. Clinical characteristics participants among presumptive TB patients at SPHMMC, Addis Ababa, Ethiopia, 2019.**

| Variables/ Characteristics | | Number of participants | Percentages (%) |
|---|---|---|---|
| Reason for diagnosis | Presumptive TB | 422 | 97 |
| | Presumptive DR-TB | 14 | 3 |
| BCG vaccination | Vaccinated | 156 | 36 |
| | Non-Vaccinated | 280 | 64 |
| TB contact history | Yes | 33 | 8 |
| | No | 403 | 92 |
| Alcohol drinking | Yes | 68 | 16 |
| | No | 368 | 84 |
| Cigarette smoking | Smokers | 22 | 5 |
| | Non-smokers | 414 | 95 |
| Night sweating | Yes | 310 | 71 |
| | No | 126 | 29 |
| Presence of fever | Yes | 318 | 73 |
| | No | 118 | 27 |
| Weight loss | Yes | 200 | 46 |
| | No | 236 | 54 |
| Presence of Cough | Yes | 364 | 83 |
| | No | 72 | 17 |
| Loss of appetite | Yes | 285 | 65 |
| | No | 151 | 35 |
| Presence of chest pain | Yes | 206 | 47 |
| | No | 230 | 53 |
| Presence of diarrhea | Yes | 57 | 13 |
| | No | 379 | 87 |
| Presence of dyspnea | Yes | 140 | 32 |
| | No | 296 | 68 |
| External-adenopathy | Yes | 63 | 14 |
| | No | 373 | 86 |
| Anti-TB Treatment | Previously treated | 110 | 25 |
| | Previously untreated | 326 | 75 |
| Presumptive DR-TB | New | 384 | 88 |
| | Relapse | 46 | 11 |
| | Failure | 6 | 1 |
| HIV status | Positive | 130 | 30 |
| | Negative | 306 | 70 |
| Tuberculosis type | PTB | 373 | 86 |
| | EPTB | 63 | 14 |
| CD4$^+$ count | <200 cells/mm$^3$ | 16 | 16 |
| | 200-350/mm$^3$ | 34 | 33 |
| | >350/mm$^3$ | 53 | 51 |
| HIV viral load | <1000/ mm$^3$ | 29 | 24 |
| | ≥1000/ mm$^3$ | 90 | 76 |

BCG = Bacillus Calmette–Guérin, DRTB = Drug Resistance Tuberculosis, EPTB = Extra Pulmonary Tuberculosis HIV = Human Immunodeficiency Virus, MDR-TB = Multidrug-Resistant Tuberculosis, MTB = Mycobacterium Tuberculosis, PTB = Pulmonary Tuberculosis

**Table 3. Socio-demographic factor analysis of *M. Tuberculosis* among presumptive TB patients at SPHMMC, Addis Ababa, Ethiopia, 2019.**

| Variables/ characteristics | | #M.TB Not detected (%) | #M.TB Detected (%) | #Total (%) | COR (95% CI) | P-value |
|---|---|---|---|---|---|---|
| Sex | Male | 208 (93) | 16 (7) | 224 (51) | 1.4(0.6–3.1) | 0.4 |
| | Female | 201 (95) | 11 (5) | 212 (49) | 1 | |
| Age groups | <15 years | 36 (92) | 3 (8) | 39 (9) | 1.8(0.4, 8.1) | 0.41 |
| | 15–24 years | 52 (93) | 4 (7) | 56 (13) | 1.7(0.4, 6.6) | 0.44 |
| | 25–34 years | 92 (94) | 6 (6) | 98 (22) | 1.5(0.4, 4.9) | 0.55 |
| | 35–49 years | 118 (93) | 9 (7) | 127 (29) | 1.7(0.6, 5.2) | 0.36 |
| | >50 years | 111(96) | 5 (4) | 116 (27) | 1 | |
| Residence | Urban | 227 (95) | 13 (5) | 240 (55) | 1 | |
| | Rural | 182 (93) | 14 (7) | 196 (45) | 1.4(0.6, 2.9) | 0.4 |
| Family size/house | 1–3 | 144 (95) | 8 (5) | 152 (35) | 1 | |
| | 4–6 | 204 (93) | 16 (7) | 220 (50) | 1.4(0.6, 3.4) | 0.4 |
| | >6 | 61 (95) | 3 (5) | 64 (15) | 0.9(0.3, 3.5) | 0.8 |
| Marital status | Single | 136 (93) | 10 (7) | 146 (33) | 1 | |
| | Married | 226 (95) | 12 (5) | 238 (55) | 1(0.2, 5.3) | 0.9 |
| | Divorced | 17 (85) | 3 (5) | 20 (5) | 0.8(0.2, 3.7) | 0.7 |
| | Widowed | 30 (94) | 2(9) | 32 (7) | 2.6(0.4, 17) | 0.3 |
| Occupational status | Laborer | 89 (92) | 8 (8) | 97 (22) | 1.4(0.4, 4.6) | 0.6 |
| | Government workers | 91(94) | 6 (6) | 97 (22) | 1.8(0.6, 5.9) | 0.3 |
| | Private workers | 58 (92) | 5 (8) | 63 (15) | 1.7(0.4, 6.4) | 0.4 |
| | Student | 67 (96) | 3 (4) | 70 (16) | 0.9(0.2, 4.0) | 0.9 |
| | House wife | 104 (95) | 5 (5) | 109 (25) | 1 | |
| Educational status | Illiterate | 112 (94) | 7 (6) | 119(72) | 0.7(0.3, 2.4) | 0.6 |
| | 1-8th grades | 140 (95) | 7 (5) | 147(34) | 0.6(0.2, 1.9) | 0.4 |
| | 9-12th grades | 98 (92) | 8 (8) | 106(24) | 0.9(0.3, 3.0) | 0.9 |
| | >12th grade | 59 (92) | 5 (8) | 64 (15) | 1 | |
| Monthly Income | <100 Birr | 56 (93) | 4 (7) | 60 (14) | 1.4(0.2, 13) | 0.8 |
| | 100–1000 Birr | 79 (95) | 4 (5) | 83 (19) | 1(0.1, 9) | 0.9 |
| | 1001–2000 Birr | 147 (95) | 8 (5) | 155(36) | 1(0.3, 9) | 0.9 |
| | 2001–3000 Birr | 54 (92) | 5 (8) | 59 (14) | 1.8(0.2, 16) | 0.6 |
| | 3001–4000 Birr | 32 (100) | 0 () | 32 (7) | - | 0.9 |
| | 4001–5000 Birr | 20 (80) | 5 (20) | 25 (6) | 5(0.5, 46) | 0.2 |
| | >5001 Birr | 20 (91) | 2 (9) | 22 (5) | 1 | |

N.B: BCG = Bacillus Calmette–Guérin, COR = Crude Odds Ratio, DRTB = Drug Resistance Tuberculosis, HIV = Human Immunodeficiency Virus,

MDR-TB = Multidrug-Resistant Tuberculosis, MTB = Mycobacterium Tuberculosis, TB = Tuberculosis

(4.5%), cough 24/436 (5.5%), loss of appetite 20/436 (4.5%), and chest pain 16/436 (3.7%). The lowest results were reported in those patients with dyspnea 9/436 (2.0%), diarrhea 3/436 (0.7%), and palpable lymphadenopathy 3/436 (0.7%).

Although tuberculosis has been frequently observed in people who have had a history of contact with TB infected person who has already developed a cough, as well as cigarette smokers and alcohol users, our data suggest that these people are not significantly more likely to develop tuberculosis. These findings were different from the studies done in Addis Ababa, Ethiopia in 2011 and north Gondar in 2015 [13, 14].

The possible reason could be the lower number of participants diagnosed with presumptive DRTB and the fact that most participants live in an urban area. Higher results were observed

**Table 4. Clinical factors associated with magnitude of *M. tuberculosis* among presumptive TB patients at SPHMMC, Addis Ababa, Ethiopia, 2019.**

| Variables/ characteristics | | #M. TB not detected (%) | #M.TB detected (%) | #Total (%) | COR (95% CI) | P-value |
|---|---|---|---|---|---|---|
| Reason for Diagnosis | Presumptive TB | 397 (94) | 25 (6) | 422 (97) | 1 | |
| | Presumptive DRTB | 12 (86) | 2 (14) | 14 (3) | 2.6(0.6, 12) | 0.2 |
| BCG Vaccination | Vaccinated | 147 (94) | 9 (6) | 156 (36) | 1 | |
| | Non-Vaccinated | 262 (94) | 18 (6) | 280 (64) | 1.1(0.5, 2.6) | 0.7 |
| TB contact History | Yes | 28 (85) | 5 (15) | 33 (8) | 3.1(1.1, 8.7) | 0.03 |
| | No | 381 (95) | 22 (5) | 403 (92) | 1 | |
| Alcohol Drinking | Yes | 62 (91) | 6 (9) | 68 (16) | 1.6(0.6, 4.1) | 0.3 |
| | No | 347(94) | 21(6) | 368 (84) | 1 | |
| Cigarette smoking | Smokers | 20 (91) | 2 (9) | 22 (5) | 1.6(0.3, 7.0) | 0.5 |
| | Non-smokers | 389 (94) | 25 (6) | 414 (95) | 1 | |
| Chest X-ray | Pneumonia | 25 (89) | 3 (11) | 28 (7) | 3(33, 319) | 0.02 |
| | Interstitial | 28 (90) | 3 (10) | 31(7) | 3(0.3, 30) | 1.0 |
| | Bronchiectasis | 11(92) | 1 (8) | 12 (3) | 2.6(0.3, 27) | 0.34 |
| | Bilateral | 6 (43) | 8 (57) | 14 (3) | 9(0.9, 8) | 0.4 |
| | Unilateral | 14 (74) | 5 (24) | 19 (4) | 0.5(0.6, 4.3) | 0.5 |
| | Normal | 324 (98) | 7 (2) | 331(76) | 1 | |
| Anti-TB treatment | Untreated | 103 (94) | 7(6) | 110 (25) | 1 | |
| | Previously treated | 306 (94) | 20 (6) | 326 (75) | 1.1(0.4, 2.5) | 0.9 |
| Presumptive DRTB | New | 362 (94) | 24 (6) | 386 (89) | 1 | |
| | Relapse | 44 (91) | 2 (9) | 46 (11) | 0.7(0.2, 3) | 0.6 |
| | Failure | 3 (75) | 1(25) | 4 (1) | 5.0(0.5, 5.0) | 0.2 |
| HIV status | Positive | 120 (92) | 10 (8) | 130 (30) | 1.4(0.6, 3.1) | 0.4 |
| | Negative | 289 (94) | 17 (6) | 306 (70) | 1 | |
| CD4 count/ mm$^3$ blood | <200 | 16 (100) | 0 (0) | 16 (15) | 1.2(0.9, 2.4) | 0.9 |
| | 200–350 | 29 (85) | 5 (15) | 34 (33) | 8.9(0.5, 0.9) | 0.049 |
| | ≥350 | 52 (96) | 2 (4) | 54 (52) | 1 | |
| Viral Load /mm$^3$ blood | <1000 | 27 (93) | 2 (7) | 29 (24) | 1 | |
| | ≥1000 | 84 (93) | 6 (7) | 90 (76) | 0.9(0.2, 5.0) | 0.9 |

BCG = Bacillus Calmette–Guérin, CD = Cluster of Differentiation, Crude Odds Ratio, DRTB = Drug Resistance Tuberculosis, HIV = Human Immunodeficiency Virus, MDR-TB = Multidrug-Resistant Tuberculosis, MTB = Mycobacterium Tuberculosis, TB = Tuberculosis

in patients who had previously been treated with anti-TB medications (20/436, or 4.5%), as well as new patients with a presumptive diagnosis of drug-resistant tuberculosis 24/436 (5.5%).

There was a statistically significant link between culture-positive pulmonary tuberculosis and TB contact history, pneumonia, and CD 4+ counts, as well as several tuberculosis patient symptoms such as weight loss. The earlier study also found a link between pulmonary tuberculosis and the number of CD4+ cells in HIV patients and the amount of virus in their blood [13, 15].

The current result seems similar with reports of study conducted in Addis Ababa, Ethiopia in 2017 [16], prisons settings of East Gojjam Zone, Northwest Ethiopia using GeneXpert MTB/RIF, 9(3.4%) [17] and 9.9% of the study conducted in extra pulmonary tuberculosis at University of Gondar, Northwest Ethiopia [18]. This overall culture confirmed *M.tuberculosis*, 27/436(6.2%) magnitude is lower than the study conducted in the Health Centers of Addis Ababa, Ethiopia reported as 46.0% (233/506) [13], from Metehara sugar factory hospital, eastern Ethiopia (14.2%) and 124 (32.2%) of studied in two public hospitals in East Gojjam zone, northwest Ethiopia [19].

We detected a reduced prevalence of tuberculosis (24.6%) when compared to a retrospective study report from the University of Gondar Hospital from January 2013 to August 2015 [20]. Our results were also lower than those of a study conducted in Debre Markos Referral Hospital in Ethiopia, which found a prevalence of 23.2% utilizing the GeneXpert MTB/RIF assay.

The difference could be due to the different diagnostic methods we used; for example, in our cases, we used the sputum sedimentation concentration technique for microscopic smear examination, GeneXpert MTB/RIF assay, and finally, LJ culture for confirmation, whereas in previous studies, a single diagnostic tool was used, such as stained by Ziehl-Neelsen staining and examined by Microscopy in Metehara [18], GeneXpert MTB/RIF in prisons settings of East Gojjam Zone [17]. This low prevalence could also indicate that TB infection control in our study area, Addis Ababa, Ethiopia, is relatively good.

From the overall confirmed *M. tuberculosis* 6.2% (27/436), a total of three *M. tuberculosis* strain showed resistance pattern to anti-tuberculosis drug, of which two of them were multi drug (INH and RIF) resistance strains. This result was lower than the study conducted in the University of Gondar Hospital, northwest Ethiopia which was reported as 71(15%) of tuberculosis-presumptive cases were resistant to rifampicin [20], and 15.58% of two public hospitals in East Gojjam zone, northwest Ethiopia [19], and 12 (10.3%) patients referred to Debre markos Referral Hospital, Ethiopia [21].

Only 10(7.7%) of the 130 HIV-infected patients tested positive for MTB, one mono (INH) resistant and one MDR-TB (INH+RIF) resistant strain were found in this seropositive figure.

In terms of viral load and tuberculosis, only one mono resistant strain was discovered in the participant serum, which has a high viral load count (1000/mmm3). This could be linked to HIV infection, which causes anti-TB drug mal-absorption and immunological suppression, leading to resistance, and our findings are backed up by previous research [22, 23].

According to the bivariate logistic analysis, patients with a presumptive diagnosis of drug resistance were two times more likely 2.6 times (95% CI 0.6, 12, p = 0.2) to acquire tuberculosis than those with a presumptive diagnosis of tuberculosis. Patients who also had night sweating were two times more likely to get tuberculosis 2.4 times (95% CI 0.8, 7.2, p = 0.1) than those who did not have. When compared to patients who did not have chest pain, having chest pain was also associated with had 1.6 times, (95%t CI 0.8, 3.7, p = 0.2) greater risk of getting *Mycobacterium tuberculosis*.

## Conclusion

In general, this study found low-magnitude *M. tuberculosis* in patients with presumptive diagnosis of TB at SPHMMC in Addis Ababa, Ethiopia. And three DRTB strains, including two MDR strains, were discovered in individuals with a history of failure, relapse, and previous anti-TB treatment.

Contact with tuberculosis-infected patients, weight loss, pneumonia on radiological examination, and low CD4+ levels were all found to be linked with *M. tuberculosis*. To maintain this low illness outcome, health education on tuberculosis, TB control programs, and large community-based studies should be continued. To lessen the incidence of MDR-TB, it is also advised that TB infection control activities be strengthened and DOT be properly implemented.

## Supporting information

**S1 File. This is the S1 of English and Amharic language version of the questionnaire.** (DOCX)

## Acknowledgments

The authors would like to express their gratitude to the staff of the Departments of Medical Laboratory Science, College of Health Sciences, Addis Ababa University, and the Department of National Tuberculosis Reference Laboratory, as well as the management of St. Paul's Hospital Millennium Medical College (SPHMMC). Finally, we thank all study participants. Last but not least, authors appreciated Dr. Yoseph Solomon Bezabih and Dr. Bereket Fantahun for their contribution of revising English language and maximizing the readability of the manuscript.

## Author Contributions

**Conceptualization:** Melkayehu Kassa, Kassu Desta, Rozina Ambachew, Zenebe Gebreyohannes, Alganesh Gebreyohanns, Nuhamen Zena, Misikir Amare, Betselot Zerihun, Melak Getu, Addisu Gize.

**Data curation:** Melkayehu Kassa, Kassu Desta, Rozina Ambachew, Zenebe Gebreyohannes, Alganesh Gebreyohanns, Nuhamen Zena, Misikir Amare, Betselot Zerihun, Melak Getu, Addisu Gize.

**Formal analysis:** Kassu Desta, Rozina Ambachew, Zenebe Gebreyohannes, Alganesh Gebreyohanns, Nuhamen Zena, Misikir Amare, Betselot Zerihun, Melak Getu, Addisu Gize.

**Funding acquisition:** Melkayehu Kassa.

**Investigation:** Melkayehu Kassa, Misikir Amare, Betselot Zerihun, Melak Getu.

**Methodology:** Melkayehu Kassa.

**Supervision:** Melkayehu Kassa.

**Writing – original draft:** Melkayehu Kassa, Kassu Desta, Addisu Gize.

**Writing – review & editing:** Addisu Gize.

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
