## [Decision Letter · Decision Letter 0]

16 Mar 2022

PONE-D-22-03876Magnitude of Multidrug Resistance Mycobacterium tuberculosis and associated factors among presumptive patients at St. Paul’s Hospital Millennium Medical College, Addis Ababa, EthiopiaPLOS ONE

Dear Dr. Yeshanew,

Thank you for submitting your manuscript to PLOS ONE. After careful consideration, we feel that it has merit but does not fully meet PLOS ONE’s publication criteria as it currently stands. Therefore, we invite you to submit a revised version of the manuscript that addresses the points raised during the review process.

Please submit your revised manuscript. If you will need significantly more time to complete your revisions, please reply to this message or contact the journal office at plosone@plos.org. Please include the following items when submitting your revised manuscript:A rebuttal letter that responds to each point raised by the academic editor and reviewer(s). You should upload this letter as a separate file labeled 'Response to Reviewers'.A marked-up copy of your manuscript that highlights changes made to the original version. You should upload this as a separate file labeled 'Revised Manuscript with Track Changes'.An unmarked version of your revised paper without tracked changes. You should upload this as a separate file labeled 'Manuscript'.

We look forward to receiving your revised manuscript.

Kind regards,

Frederick Quinn

Academic Editor

PLOS ONE

Journal Requirements:

2. Thank you for submitting the above manuscript to PLOS ONE. During our internal evaluation of the manuscript, we found significant text overlap between your submission and the following previously published works, some of which you are an author.

- https://www.hindawi.com/journals/trt/2016/6207457/

- https://www.dovepress.com/rifampicin-resistant-mycobacterium-tuberculosis-among-tuberculosis-pre-peer-reviewed-fulltext-article-IDR

- https://www.ghspjournal.org/content/1/1/18.full

- https://www.ncbi.nlm.nih.gov/pmc/articles/PMC3557681/

Please revise the manuscript to rephrase the duplicated text, cite your sources, and provide details as to how the current manuscript advances on previous work. Please note that further consideration is dependent on the submission of a manuscript that addresses these concerns about the overlap in text with published work.

"Authors would like to thank the Departments of Medical Laboratory Science, College of Health

Sciences, Addis Ababa University, Department of national tuberculosis reference laboratory

staffs, members of Ethiopian Public Health Institute who were willing to use laboratory space

and consumable fund, and also St. Paul’s Hospital Millennium Medical College (SPHMMC)

management, last but not lease we are grateful for all study participants."

"This research work was supported by Addis Ababa University, Ethiopia. 

Reviewers' comments:

Reviewer's Responses to Questions

**Comments to the Author**

1. Is the manuscript technically sound, and do the data support the conclusions?

Reviewer #1: Partly

Reviewer #2: Partly

2. Has the statistical analysis been performed appropriately and rigorously? 

Reviewer #1: No

Reviewer #2: I Don't Know

3. Have the authors made all data underlying the findings in their manuscript fully available?

Reviewer #1: Yes

Reviewer #2: Yes

4. Is the manuscript presented in an intelligible fashion and written in standard English?

Reviewer #1: No

Reviewer #2: No

5. Review Comments to the Author

Reviewer #1: his deals on an important and current issue, TB and drug resistance TB in Ethiopia. However, the manuscript poorly organized and need major revision. I try to mentioned some points on the main manuscript but many more left .

Reviewer #2: Comments

The author picked an interesting topic that has public health importance but it is not a new topic. Such kind of work has important role in monitoring the distribution of MDR-TB in the study area. The study aimed to determine the magnitude of MDR-TB and associated factors as it is seen from the title of the manuscript and objectives mentioned in the abstract; however, in the background, results, and discussion there is a different scenario. There are concerns I encountered in this manuscript some of them are: what was the aim of this study? Why it vary from one section to the other? There are inconsistencies in this regard that can affect the results, discussion, and conclusion section. The other one is the sample size issue, unless it is resolved, I think it will affect the manuscript. Authors are advised to carefully revise the manuscript. My comments are depicted below.

I. General comments

1. Revise language

2. Follow the scientific way of writing to name of bacteria

3. There are several typographical errors that need revision for example in citing a reference in one place it is placed in [] in some other place it is presented as superscript. Sometimes reference which is not on the list is cited.

4. Proper use of abbreviations: If abbreviations appear for the first time in the manuscript use the expanded form then after use the shortened form.

5. I suggest the use of ‘GeneXpert’ instead of “X-pert MTB/RIF”

II. Abstract

1. Several typos example “….from Jan to July 2019…” change Jan to January

2. The title does not reflect the objective (prevalence of MTB vs Prevalence of MDR; factors associated with the magnitude of MTB? Or MDR? ….justify or modify the title

3. The background in the abstract does not reflect the title

4. Which method was used to detect MDR? The method in the abstract needs revision it should go with the title and objective (detection of MDR) what was the need for microscopy?

5. What was the sample size of this study? 422? 436?

6. Omit socio-demographic data, focus on the main findings: MDR, factors associated with MDR (this was not shown anywhere in the manuscript)

7. A statement “Out of the total participants, the overall confirmed Mycobacterium

tuberculosis was through X-pert MTB/RIF assay and LJ culture media was 27 (6.2%), and three isolates were resistant for either INH or RIF drug, while two of them were MDR-TB based on line probe assays method” needs revision

8. In a statement “Previous TB-contact history, patient weight loss, having pneumonia

with chest X-ray finding, and CD4+ T-cells count 200-350/mm3 of blood were significantly associated predictors for MTB infection” does not go with the title and objective. As I understand the aim of the study is to determine factors associated with MDR-TB

9. What is the prevalence of MDR and its predictors, these two should be given emphasis in results and conclusion.

III. Background

1. The first sentence “…..caused by strains belonging to …” as strains is inappropriately used, modify as “…caused by Mycobacterium tuberculosis complex…”

i. In a statement “Globally, the estimated prevalence of MDR-TB was 3.3% in newly diagnosed patients in the WHO 2015 report. This was higher to 20% in patients with a history of anti-TB treatment(30).” Reference # 30 does not exist in the reference list.

2. Use updated information in the background: 2017 WHO report is old

3. Also use the updated information for Ethiopia instead of using 2005 report

4. The last sentence of the background is not in line with what has been mentioned in the title and abstract. They contradict each other, justify or revise

a. “Therefore, the goal of this study was to determine magnitude of Mycobacterium tuberculosis and its associated factors among TB- presumptive patients referred to St. Paul’s Hospital Millennium Medical College, Addis Ababa, Ethiopia.”

b. “Therefore, the goal of this study was to determine magnitude of Multi Drug Mycobacterium tuberculosis (MDR-TB) and its associated factors among TB- presumptive patients at St. Paul’s Hospital Millennium Medical College, Addis Ababa, Ethiopia”

5. Use another word instead of the “the goal of this study…”

IV. Materials and Methods

1. In the study area, mention the TB clinic (patient flow, service…etc)

2. What does ‘Jan’ stands for? Is it January?

3. Study variables should go with the title and objective of the study: MDR-TB and predictors of MDR-TB

4. Determination of sample size is not clear. If p=23% (reference is not given, it is not clear whether it is the prevalence of TB or MDR), d=5% is used the sample size will be 272 + 10% (n=299) which is different from what is mentioned in the manuscript (n=422). Moreover, a sample size different from these two was used in the result section (n=436).

5. Mention sampling technique; non-probability sampling technique (convenient)? Systematic random sampling technique? Random sampling technique?

6. Detail is needed on data collection procedure: how the questionnaire was validated? Move clinical specimen collection to the laboratory section.

7. How the sample was collected from children who are unable to provide sputum? I think it is difficult to obtain sputum from children, how did you handle it?

8. In Laboratory procedures section mention what was done to identify MTB, MDR (microscopy?, culture, GeneXpert) instead of description or performance of the methods.

9. I am not sure if the use of trademark is allowed by the journal (®), check it out.

10. Data quality assurance: mention how the quality of data was maintained both for socio-demographic, clinical characteristics, and Laboratory data

11. Data analysis and interpretation: what does data editing mean? You should have considered multivariate analysis.

12. In a statement “…to assess the association between different factors..” what are these factors?

13. Ethical consideration: correct “…Department of Medical Laboratory Sciences” as “…….Science” check if it is school or department. Does the department of Medical Laboratory Science have IRB; most of the time IRB is at the college level.

14. In the sentence “…then submitted to laboratory department” which laboratory department? And why?

15. Did you obtain assent for children?

16. Include operational definition for MDR, presumptive TB, presumptive MDR-TB

V. Results

1. The whole result is based on the sample size (436vs422vs299) which is different from the one mentioned in the method section (where do 436 come from?).

2. “…had monthly income 100-1000 Ethiopian Birr, table 1.” May be re-written as “……………Ethiopian Birr (Table 1).” Do the same wherever it applies.

3. In a clinical data: “About 422 (96.8%) of the participants were presumptive TB..” how is this different from “From the total 374 (85.8%) were suspected for pulmonary tuberculosis and 62 (14.2%) were suspected for extra-pulmonary tuberculosis”

4. Why most of the study participants are HIV positive? What kind of impact does it have on your study?

5. Rename subheading ‘Bivariate analysis’ as ‘Factors associated with MDR-TB’

6. As mentioned above, the analysis does not go with title and objective and also multivariate analysis was not conducted.

VI. Discussion

1. Because of some points mentioned above, the discussion is not in order: it should follow the main findings of the study: should discuss the magnitude of MDR that is followed by risk factors. Discussion can be re-written after addressing the comments given above.

2. Some types, page 11: “Mycobacterium Tuberculosis” correct as Mycobacterium tuberculosis, ‘East Gojjam zone, northwest Ethiopia,19” & “eastern Ethiopia (14.2%)18 “ references are in superscript put them in []

3. In a statement “The possible reason for the difference might be associated with the variation of the diagnostic methods we used, for example in our cases we used sputum sedimentation concentration technique for microscopic smear examination, Gene X-pert assay and finally LJ culture for”confirmation whereas, a single diagnostic tool used in the previous study like;stained by ZiehlNeelsen staining and examined by Microscopy in the case of Metehara [18]” in the method section you did not mention that you have used sputum sedimentation concentration technique. Moreover if you have used all this the fining should be higher.

VII. Conclusions

1. Conclusion needs revision

VIII. References and Tables

2. Reference writing style is not uniform (#1, 7, 8, 16, 18, ), Add URL for refer #3,

3. Tables: the labeling of table 1 is not correct (it is not for MDR?), you may omit Colum 3 and 4 (MTB present & MTB absent), the age category could have been reduced. Replace ‘illiterate’ with no formal education, monthly income category is not reasonable. The same comments apply for table 2 (correct labeling of the table and omit Colum 3 &4. Expand all abbreviations mentioned in the table under the tables. Re-write labeling of table 3 &4 analysis of what with what (MDR with socio or TB with socio) please check your title of manuscript and objective. And also consider multivariate analysis for variables with p<0.25. In the tables include both frequency and percentages in each cell.

6. PLOS authors have the option to publish the peer review history of their article (what does this mean?). If published, this will include your full peer review and any attached files.

Reviewer #1: No

Reviewer #2: No

---

## [Author Response · Author response to Decision Letter 0]

1 May 2022

View Letter

Date: Mar 16 2022 10:05AM

To: "Addisu Gize Yeshanew" konjoaddisu@gmail.com

From: "PLOS ONE" plosone@plos.org

Subject: PLOS ONE Decision: Revision required [PONE-D-22-03876]

 Attachment(s): PONE-D-22-03876_ suggesions and comnts.pdf 

PONE-D-22-03876

Magnitude of Multidrug Resistance Mycobacterium tuberculosis and associated factors among presumptive patients at St. Paul’s Hospital Millennium Medical College, Addis Ababa, Ethiopia

PLOS ONE

Dear Dr. Yeshanew,

Thank you for submitting your manuscript to PLOS ONE. After careful consideration, we feel that it has merit but does not fully meet PLOS ONE’s publication criteria as it currently stands. Therefore, we invite you to submit a revised version of the manuscript that addresses the points raised during the review process.

Please submit your revised manuscript. If you will need significantly more time to complete your revisions, please reply to this message or contact the journal office at plosone@plos.org. Please include the following items when submitting your revised manuscript:

We look forward to receiving your revised manuscript.

Kind regards,

Frederick Quinn

Academic Editor

PLOS ONE

Response: Dear Frederick Quinn, Academic Editor, PLOS ONE

We authors would like to thank for your invitation to submit our revised version of the manuscript, PONE-D-22-03876 titled” Magnitude of Multidrug Resistance Mycobacterium tuberculosis and associated factors among presumptive patients at St. Paul’s Hospital Millennium Medical College, Addis Ababa, Ethiopia”, that addresses the points raised during the review process.

 We included a point-by-point response within the 'Response to Reviewers' box in the submission system and modifications are highlighted in the track changes of the original manuscript, and we uploaded also the clean or unmarked version of our revised paper without tracked changes. 

Journal Requirements:

Response: Thank you for the concern, the PLOS ONE’s style requirement has been incorporated accordingly in all sections.

2. Thank you for submitting the above manuscript to PLOS ONE. During our internal evaluation of the manuscript, we found significant text overlap between your submission and the following previously published works, some of which you are an author.

- https://www.hindawi.com/journals/trt/2016/6207457/

- https://www.dovepress.com/rifampicin-resistant-mycobacterium-tuberculosis-among-tuberculosis-pre-peer-reviewed-fulltext-article-IDR

- https://www.ghspjournal.org/content/1/1/18.full

- https://www.ncbi.nlm.nih.gov/pmc/articles/PMC3557681/

Please revise the manuscript to rephrase the duplicated text, cite your sources, and provide details as to how the current manuscript advances on previous work. Please note that further consideration is dependent on the submission of a manuscript that addresses these concerns about the overlap in text with published work.

Response: Thank you for your concern again and the constructive comment. We have rephrased and completely modified each word taken from the previous published articles.

Response: Regarding our survey tool or questionnaire, we have uploaded as supporting file and any interested can replicate for the study and analyses.

"Authors would like to thank the Departments of Medical Laboratory Science, College of Health

Sciences, Addis Ababa University, Department of national tuberculosis reference laboratory

staffs, members of Ethiopian Public Health Institute who were willing to use laboratory space

and consumable fund, and also St. Paul’s Hospital Millennium Medical College (SPHMMC)

management, last but not lease we are grateful for all study participants."

"This research work was supported by Addis Ababa University, Ethiopia. 

Response: We included our Funding statement in the cover letter, same time the statement which stated about our funding information is removed from the revised manuscript.

Response: The comment is well taken and based on the comment; we have stated the ethics statement only in the method section of the revised manuscript.

Reviewers' comments:

Reviewer's Responses to Questions

Comments to the Author

1. Is the manuscript technically sound, and do the data support the conclusions?

Reviewer #1: Partly

Reviewer #2: Partly

Response: It is true that one’s perception different from other. However, we author’s believe that our manuscript technically sound, and do the data support the manuscript conclusions.

2. Has the statistical analysis been performed appropriately and rigorously? 

Reviewer #1: No

Reviewer #2: I Don't Know

Response: Again it may be due to personal difference or may be because of reviewers are not public health professionals. However, we author’s consulted by health professionals for our statistical analysis, and we believe that the statistical analysis has been performed appropriately.

3. Have the authors made all data underlying the findings in their manuscript fully available?

Reviewer #1: Yes

Reviewer #2: Yes

Response: Thank you.

4. Is the manuscript presented in an intelligible fashion and written in standard English?

Reviewer #1: No

Reviewer #2: No

Response: The comment is well taken. By now our manuscript revised detail to have intelligible fashion and Standard English language presentation to make it clear for the readers.

5. Review Comments to the Author

Reviewer #1: his deals on an important and current issue, TB and drug resistance TB in Ethiopia. However, the manuscript poorly organized and need major revision. I try to mention some points on the main manuscript but many more left.

Response: Again, this comment is well taken. Based on each comment mentioned in the main manuscript document, we have done major revision and incorporated the suggested comments on the revised manuscript.

We authors glad to appreciate reviewer#1effort for his comment/concern and questions for unclear points found in main manuscript document.

Reviewer #2: Comments

The author picked an interesting topic that has public health importance but it is not a new topic. Such kind of work has important role in monitoring the distribution of MDR-TB in the study area. The study aimed to determine the magnitude of MDR-TB and associated factors as it is seen from the title of the manuscript and objectives mentioned in the abstract; however, in the background, results, and discussion there is a different scenario. There are concerns I encountered in this manuscript some of them are: what was the aim of this study? 

Response: Thank you for your interest and stating the importance of the study as public health problem. By now, we have taken your comment and concern and define the objective of the study.

Why it vary from one section to the other? There are inconsistencies in this regard that can affect the results, discussion, and conclusion section. 

Response: Again the comment is well taken. We have modified many things in the revised manuscript to keep the consistency in result, discussion and conclusion.

The other one is the sample size issue, unless it is resolved, I think it will affect the manuscript. 

Response: With all due respect to the reviewer, we believe that this point is not correct and our sample size is calculated with single population proportion formula, and by now it is resolved in the revised document.

Authors are advised to carefully revise the manuscript. My comments are depicted below.

Response: Well come for your comments and thank you for investing your time on our manuscript.

I. General comments

1. Revise language

Response: The comment is well taken. By now our manuscript revised detail to have intelligible fashion and Standard English language presentation to make it clear for the readers.

2. Follow the scientific way of writing to name of bacteria

Response: We agree with this reviewer that very few bacteria were named non-scientifically, but now we have scientifically corrected their names.

3. There are several typographical errors that need revision for example in citing a reference in one place it is placed in [] in some other place it is presented as superscript. Sometimes reference which is not on the list is cited. 

Response: Thank you for your constructive comment. We have corrected our manuscript citation and references based on the journal requirement.

4. Proper use of abbreviations: If abbreviations appear for the first time in the manuscript use the expanded form then after use the shortened form. 

Response: Again the comment is well taken, and by now we have corrected our manuscript abbreviation. 

5. I suggest the use of ‘GeneXpert’ instead of “X-pert MTB/RIF” 

Response: Again the comment is well taken, and by now we have corrected our manuscript as per suggestion as GeneXpert. 

II. Abstract

1. Several typos example “….from Jan to July 2019…” change Jan to January

Response: Thank you. We have corrected as per the comment.

2. The title does not reflect the objective (prevalence of MTB vs Prevalence of MDR; factors associated with the magnitude of MTB? Or MDR? ….justify or modify the title

Response: The comment is accepted. Based on the comment the title modified as “Magnitude of Mycobacterium tuberculosis, Drug resistance and associated factors among presumptive patients at St. Paul’s Hospital Millennium Medical College, Addis Ababa, Ethiopia” 

3. The background in the abstract does not reflect the title 

Response: Based on the comment, now our background in the abstract section corrected to reflect the title. 

4. Which method was used to detect MDR? The method in the abstract needs revision it should go with the title and objective (detection of MDR) what was the need for microscopy? 

Response: The methods used for MDR detection were GeneXpert MTB/RIF assay and LJ culture media. We used Microscopy for cross checking, and now we have revised the abstract section.

5. What was the sample size of this study? 422? 436?

Response: It was proposed the minimum sample size using single population formula as 422. However, we used 436 participants in the actual data collection procedure (more than the determined sample size) during the study period.

6. Omit socio-demographic data, focus on the main findings: MDR, factors associated with MDR (this was not shown anywhere in the manuscript)

With all due respect to the reviewer, we believe that this point is not correct. Socio-demographic data is mandatory for magnitude of M. tuberculosis and to describe the statistical associations.

7. A statement “Out of the total participants, the overall confirmed Mycobacterium

tuberculosis was through X-pert MTB/RIF assay and LJ culture media was 27 (6.2%), and three isolates were resistant for either INH or RIF drug, while two of them were MDR-TB based on line probe assays method” needs revision

Response: Based on the comment, we have revised the stated statement.

8. In a statement “Previous TB-contact history, patient weight loss, having pneumonia

with chest X-ray finding, and CD4+ T-cells count 200-350/mm3 of blood were significantly associated predictors for MTB infection” does not go with the title and objective. As I understand the aim of the study is to determine factors associated with MDR-TB

Response: Based on the above comment, now we have revised the title to reflect exactly the objective and the associated factors.

9. What is the prevalence of MDR and its predictors, these two should be given emphasis in results and conclusion.

Response: Thank you very much. The prevalence of MDR-TB was 0.5%, and the predictors are not previous TB-contact history, patient weight loss, CD4+ T-cells count 200-350/mm3 of blood were significantly associated predictors for MTB infection, and not for MDR-TB.

III. Background

1. The first sentence “…..caused by strains belonging to …” as strains is inappropriately used, modify as “…caused by Mycobacterium tuberculosis complex…”

i. In a statement “Globally, the estimated prevalence of MDR-TB was 3.3% in newly diagnosed patients in the WHO 2015 report. This was higher to 20% in patients with a history of anti-TB treatment(30).” Reference # 30 does not exist in the reference list.

Response: The comment is well taken and raised by the first reviewer, and by now it correctly cited.

2. Use updated information in the background: 2017 WHO report is old

Response: Thank you for your suggestion and our manuscript is revised using the updated WHO 2020 report.

3. Also use the updated information for Ethiopia instead of using 2005 report

Response: Thank you for your suggestion and our manuscript is revised using the updated 2018 national study.

4. The last sentence of the background is not in line with what has been mentioned in the title and abstract. They contradict each other, justify or revise

a. “Therefore, the goal of this study was to determine magnitude of Mycobacterium tuberculosis and its associated factors among TB- presumptive patients referred to St. Paul’s Hospital Millennium Medical College, Addis Ababa, Ethiopia.”

b. “Therefore, the goal of this study was to determine magnitude of Multi Drug Mycobacterium tuberculosis (MDR-TB) and its associated factors among TB- presumptive patients at St. Paul’s Hospital Millennium Medical College, Addis Ababa, Ethiopia” 

Response: Thank you for your comment and now we have revised as suggested in the option a.

5. Use another word instead of the “the goal of this study…”

Response: We changed the goal of this study by the aim of this study.

IV. Materials and Methods

1. In the study area, mention the TB clinic (patient flow, service…etc)

Response: The comment is taken and described in the revised document.

2. What does ‘Jan’ stands for? Is it January?

Response: Yes to mean that January. Now we have corrected it as January in the revised manuscript.

3. Study variables should go with the title and objective of the study: MDR-TB and predictors of MDR-TB

Response: Now we have corrected the title and objective of the study, and both dependent and independent variables are in line with the revised title and objectives.

4. Determination of sample size is not clear. If p=23% (reference is not given, it is not clear whether it is the prevalence of TB or MDR), d=5% is used the sample size will be 272 + 10% (n=299) which is different from what is mentioned in the manuscript (n=422). Moreover, a sample size different from these two was used in the result section (n=436).

Response: The comment is accepted. Now we have revised that our sample size calculation using single population proportion formula, and the minimum sample size for the study was 422 participants. However that, during the actual collection of the study period we have collected data from 436 participants. (I think it is expected to collect more data from the pre-determined sample to be more representative). 

5. Mention sampling technique; non-probability sampling technique (convenient)? Systematic random sampling technique? Random sampling technique?

Response: We have already mentioned our sampling technique as “consecutive sampling technique was used to select the study population”, which is usually applicable to collect data from health institution as probability sampling techniques, since it is difficult to apply Systematic random sampling technique or random sampling technique. This sampling technique is considered as probability sampling method in most literatures.

 6. Detail is needed on data collection procedure: how the questionnaire was validated? Move clinical specimen collection to the laboratory section. 

Response: It is already stated as “the questionnaire was pre-tested and proper training prior to the actual data collection was given for data collectors. The necessary adjustments were made after the pre-test”, in the data quality assurance section and collection procedure is well revised and categorized as: Collection using questionnaire, laboratory collection procedures, using GeneXpert, Microscopic examination and culturing method.

7. How the sample was collected from children who are unable to provide sputum? I think it is difficult to obtain sputum from children, how did you handle it?

Response: Thank you for your concern. We were describing that taking from participants of 2-4 ml of clinical sputum samples, pus, lymph node aspirate or peritoneal, pleural fluid and gastric aspirate. Sputum sample was taken usually from children older than 10 years who can produce sputum. We used Gastric aspirate or induced sputum by physician support for children unable to provide sputum by coughing.

8. In Laboratory procedures section mention what was done to identify MTB, MDR (microscopy?, culture, GeneXpert) instead of description or performance of the methods.

Response: The comment is accepted, and the procedures are mentioned in the revised manuscript.

9. I am not sure if the use of trademark is allowed by the journal (®), check it out.

 Response: The comment is accepted and removed from the revised document.

10. Data quality assurance: mention how the quality of data was maintained both for socio-demographic, clinical characteristics, and Laboratory data 

Response: Thank you. By now we have clearly mentioned. 

11. Data analysis and interpretation: what does data editing mean? You should have considered multivariate analysis. 

Response: The comment is taken and removed unnecessary words, and regarding to the multivariable analysis consideration, we agree with the reviewer the importance of multivariate analysis in terms of controlling the compfounders. However that, no socio-demographic factors was a candidate (p<0.25) for multivariable analysis in the table 3, and for few variable that were associated in the table 4, unfortunately we did not do analysis for multivariable analysis. 

12. In a statement “…to assess the association between different factors..” what are these factors? 

Response: Again the comment is taken and corrected as “ to assess the association between TB and risk factors”

13. Ethical consideration: correct “…Department of Medical Laboratory Sciences” as “…….Science” check if it is school or department. Does the department of Medical Laboratory Science have IRB; most of the time IRB is at the college level. 

Response: It is corrected as Science. Yes, it is department not the school, and has its own IRB.

14. In the sentence “…then submitted to laboratory department” which laboratory department? And why? 

Response: I think this is clear, Laboratory department means the actual study area which data was collected, it is St. Paul’s Hospital Millennium Medical College, Microbiology laboratory department, while the study was approved by Department of Medical Laboratory Science, Addis Ababa, University.

15. Did you obtain assent for children?

 Response: Yes. Now we have stated in the ethics section of the revised manuscript.

16. Include operational definition for MDR, presumptive TB, and presumptive MDR-TB 

Response: Thank you for your constructive comments. By now we have included our operation definition for MDR TB, presumptive TB, and presumptive MDR-TB 

V. Results

1. The whole result is based on the sample size (436vs422vs299) which is different from the one mentioned in the method section (where do 436 come from?).

Response: The minimum sample size calculated or predetermined was 422, however the actual study participants were 436, and it is advised to use large sample size in any study to be more representative of the study result.

2. “…had monthly income 100-1000 Ethiopian Birr, table 1.” May be re-written as “……………Ethiopian Birr (Table 1).” Do the same wherever it applies.

Response: The comment is taken and modified based on the comment.

3. In a clinical data: “About 422 (96.8%) of the participants were presumptive TB..” how is this different from “From the total 374 (85.8%) were suspected for pulmonary tuberculosis and 62 (14.2%) were suspected for extra-pulmonary tuberculosis” 

Response: No difference in the interpretation except different wording. By now it is corrected be consistent as presumptive TB.

4. Why most of the study participants are HIV positive? What kind of impact does it have on your study?

Response: Of the total 130 (30%) were HIV positive participants, of which104 (81%) were on anti-HIV treatment. These may have an impact to be infected with MTB or the chance to develop TB or MDR-TB.

5. Rename subheading ‘Bivariate analysis’ as ‘Factors associated with MDR-TB’

Response: The comment is accepted and modified as reviewer’s wish bivariate analysis to factor associated to MTB.

6. As mentioned above, the analysis does not go with title and objective and also multivariate analysis was not conducted. 

Response: The comment was taken from the previous concern and suggestion. By now the title is modified as “Magnitude of Mycobacterium tuberculosis, Drug resistance and associated factors among presumptive patients at St. Paul’s Hospital Millennium Medical College, Addis Ababa, Ethiopia”, which is in line with the objective and the analysis of the manuscript.

VI. Discussion

1. Because of some points mentioned above, the discussion is not in order: it should follow the main findings of the study: should discuss the magnitude of MDR that is followed by risk factors. Discussion can be re-written after addressing the comments given above.

Response: The comment is taken and our manuscript is revised based on the comment.

2. Some types, page 11: “Mycobacterium Tuberculosis” correct as Mycobacterium tuberculosis, ‘East Gojjam zone, northwest Ethiopia,19” & “eastern Ethiopia (14.2%)18 “ references are in superscript put them in []

Response: Thank you. By now it is correctly cited as [18].

3. In a statement “The possible reason for the difference might be associated with the variation of the diagnostic methods we used, for example in our cases we used sputum sedimentation concentration technique for microscopic smear examination, Gene X-pert assay and finally LJ culture for”confirmation whereas, a single diagnostic tool used in the previous study like;stained by ZiehlNeelsen staining and examined by Microscopy in the case of Metehara [18]” in the method section you did not mention that you have used sputum sedimentation concentration technique. Moreover if you have used all this the fining should be higher.

Response: Thank you very much again. Now it is clearly stated in the method section of the revised manuscript about sputum sedimentation concentration technique. In addition the overall result of the study will be lower because of minimizing of false positive using parallel different diagnostic and confirmatory methods.

VII. Conclusions

1. Conclusion needs revision

Response: The constructive comment is well taken, and we have revised the conclusion section.

VIII. References and Tables

2. Reference writing style is not uniform (#1, 7, 8, 16, 18, ), Add URL for refer #3,

Response: By now, we have made. All the references are uniform which have URL and PubMed citation format with their own doi and PMID in the revised manuscript.

3. Tables: the labeling of table 1 is not correct (it is not for MDR?), you may omit Colum 3 and 4 (MTB present & MTB absent), the age category could have been reduced. Replace ‘illiterate’ with no formal education, monthly income category is not reasonable. 

Response: We have revised the table title and its content as per the comment.

The same comments apply for table 2 (correct labeling of the table and omit Colum 3 &4. 

Response: We have revised the table title and its content as per the comment.

Expand all abbreviations mentioned in the table under the tables. Re-write labeling of table 3 &4 analysis of what with what (MDR with socio or TB with socio) please check your title of manuscript and objective. 

Response: We have revised the table title and its content as per the comment.

And also consider multivariate analysis for variables with p<0.25. In the tables include both frequency and percentages in each cell.

Response: Regarding to the multivariable analysis consideration, we agree with the reviewer the importance of multivariate analysis in terms of controlling the compfounders. However that, no socio-demographic factors was a candidate (p<0.25) for multivariable analysis from table 3, and for few variable that were associated in the table 4, unfortunately we did not do analysis for multivariable analysis.________________________________________

6. PLOS authors have the option to publish the peer review history of their article (what does this mean?). If published, this will include your full peer review and any attached files.

Do you want your identity to be public for this peer review? For information about this choice, including consent withdrawal, please see our Privacy Policy.

Reviewer #1: No

Reviewer #2: No

 Response: No problem.

While revising your submission, please upload your figure files to the Preflight Analysis and Conversion Engine (PACE) digital diagnostic tool, https://pacev2.apexcovantage.com/. PACE helps ensure that figures meet PLOS requirements. To use PACE, you must first register as a user. Registration is free. Then, login and navigate to the UPLOAD tab, where you will find detailed instructions on how to use the tool. If you encounter any issues or have any questions when using PACE, please email PLOS at figures@plos.org. Please note that Supporting Information files do not need this step.________________________________________

---

## [Decision Letter · Decision Letter 1]

7 Jun 2022

PONE-D-22-03876R1Magnitude of Mycobacterium tuberculosis, drug resistance and associated factors among presumptive patients at St. Paul’s Hospital Millennium Medical College, Addis Ababa, EthiopiaPLOS ONE

Dear Dr. Yeshanew,

Thank you for submitting your manuscript to PLOS ONE. After careful consideration, we feel that it has merit but does not fully meet PLOS ONE’s publication criteria as it currently stands. Therefore, we invite you to submit a revised version of the manuscript that addresses the points raised during the review process. Please submit your revised manuscript. If you will need significantly more time to complete your revisions, please reply to this message or contact the journal office at plosone@plos.org. Please include the following items when submitting your revised manuscript:A rebuttal letter that responds to each point raised by the academic editor and reviewer(s). You should upload this letter as a separate file labeled 'Response to Reviewers'.A marked-up copy of your manuscript that highlights changes made to the original version. You should upload this as a separate file labeled 'Revised Manuscript with Track Changes'.An unmarked version of your revised paper without tracked changes. You should upload this as a separate file labeled 'Manuscript'.If applicable, we recommend that you deposit your laboratory protocols in protocols.io to enhance the reproducibility of your results. Protocols.io assigns your protocol its own identifier (DOI) so that it can be cited independently in the future. For instructions see: https://journals.plos.org/plosone/s/submission-guidelines#loc-laboratory-protocols. Additionally, PLOS ONE offers an option for publishing peer-reviewed Lab Protocol articles, which describe protocols hosted on protocols.io. Read more information on sharing protocols at https://plos.org/protocols?utm_medium=editorial-email&utm_source=authorletters&utm_campaign=protocols.

We look forward to receiving your revised manuscript.

Kind regards,

Frederick Quinn

Academic Editor

PLOS ONE

Journal Requirements:

Reviewers' comments:

Reviewer's Responses to Questions

**Comments to the Author**

1. If the authors have adequately addressed your comments raised in a previous round of review and you feel that this manuscript is now acceptable for publication, you may indicate that here to bypass the “Comments to the Author” section, enter your conflict of interest statement in the “Confidential to Editor” section, and submit your "Accept" recommendation.

Reviewer #1: (No Response)

Reviewer #2: All comments have been addressed

2. Is the manuscript technically sound, and do the data support the conclusions?

Reviewer #1: Partly

Reviewer #2: Yes

3. Has the statistical analysis been performed appropriately and rigorously? 

Reviewer #1: Yes

Reviewer #2: Yes

4. Have the authors made all data underlying the findings in their manuscript fully available?

Reviewer #1: No

Reviewer #2: Yes

5. Is the manuscript presented in an intelligible fashion and written in standard English?

Reviewer #1: No

Reviewer #2: Yes

6. Review Comments to the Author

Reviewer #1: Despite the manuscript generally has showed good improvement, but it has many flaw for the publication. It need additional work to improve it. I tried to identity some of them and tried to highlight on the main manuscript.

Reviewer #2: All my comments are addressed.I don't have additional comments. The manuscript is improved considerebely.

7. PLOS authors have the option to publish the peer review history of their article (what does this mean?). If published, this will include your full peer review and any attached files.

Reviewer #1: No

Reviewer #2: No

---

## [Author Response · Author response to Decision Letter 1]

23 Jun 2022

All issues raised by reviewers and editors are addressed in the revised version of the manuscript. We authors are advised to check files labelled as Response to reviewers, Manuscript with track changes, Clean Manuscript and Cover letter. 

Thank you for all your concern.

---

## [Decision Letter · Decision Letter 2]

20 Jul 2022

Magnitude of Mycobacterium tuberculosis, drug resistance and associated factors among presumptive tuberculosis patients at St. Paul’s Hospital Millennium Medical College, Addis Ababa, Ethiopia

PONE-D-22-03876R2

Dear Dr. Yeshanew,

We’re pleased to inform you that your manuscript has been judged scientifically suitable for publication and will be formally accepted for publication once it meets all outstanding technical requirements.

Kind regards,

Frederick Quinn

Academic Editor

PLOS ONE

Additional Editor Comments (optional):

Reviewers' comments:

Reviewer's Responses to Questions

**Comments to the Author**

1. If the authors have adequately addressed your comments raised in a previous round of review and you feel that this manuscript is now acceptable for publication, you may indicate that here to bypass the “Comments to the Author” section, enter your conflict of interest statement in the “Confidential to Editor” section, and submit your "Accept" recommendation.

Reviewer #1: All comments have been addressed

Reviewer #2: All comments have been addressed

2. Is the manuscript technically sound, and do the data support the conclusions?

Reviewer #1: Yes

Reviewer #2: Yes

3. Has the statistical analysis been performed appropriately and rigorously? 

Reviewer #1: Yes

Reviewer #2: Yes

4. Have the authors made all data underlying the findings in their manuscript fully available?

Reviewer #1: Yes

Reviewer #2: Yes

5. Is the manuscript presented in an intelligible fashion and written in standard English?

Reviewer #1: No

Reviewer #2: Yes

6. Review Comments to the Author

Reviewer #1: Thank you for addressing all my comments and suggestion, I previously mentioned. The manuscript is at the current stage if fine and looks organized. However, the grammar still needs thorough editing and rephrasing of some of the words

Reviewer #2: Authors have included or responded all my comments. I am satisfied with it. The English has also improved, I don’t have additional comments.

7. PLOS authors have the option to publish the peer review history of their article (what does this mean?). If published, this will include your full peer review and any attached files.

Reviewer #1: No

Reviewer #2: No

---

## [Editor Report · Acceptance letter]

22 Jul 2022

PONE-D-22-03876R2 

Magnitude of *Mycobacterium tuberculosis*, drug resistance and associated factors among presumptive tuberculosis patients at St. Paul’s Hospital Millennium Medical College, Addis Ababa, Ethiopia 

Dear Dr. Yeshanew:

I'm pleased to inform you that your manuscript has been deemed suitable for publication in PLOS ONE. Congratulations! Your manuscript is now with our production department. 

Kind regards, 

on behalf of

Dr. Frederick Quinn 

Academic Editor

PLOS ONE